

# A model of population dynamics with complex household structure and mobility: implications for transmission and control of communicable diseases

Rebecca H. Chisholm[1,2], Bradley Crammond[2], Yue Wu[3],
Asha C. Bowen[3,4], Patricia T. Campbell[2,5], Steven Y.C. Tong[6,7,8],
Jodie McVernon[2,5] and Nicholas Geard[5,9]

[1] Department of Mathematics and Statistics, La Trobe University, Bundoora, VIC, Australia
[2] Centre for Epidemiology and Biostatistics, Melbourne School of Population and Global Health, The University of Melbourne, Melbourne, VIC, Australia
[3] Wesfarmers Centre for Vaccines and Infectious Diseases, Telethon Kids Institute, Perth, WA, Australia
[4] Infectious Diseases Department, Perth Children's Hospital, Perth, WA, Australia
[5] Victorian Infectious Diseases Reference Laboratory Epidemiology, University of Melbourne at the Peter Doherty Institute for Infection and Immunity, Melbourne, VIC, Australia
[6] Doherty Department, University of Melbourne at the Peter Doherty Institute for Infection and Immunity, Melbourne, VIC, Australia
[7] Victorian Infectious Diseases Service, The Royal Melbourne Hospital, Melbourne, VIC, Australia
[8] Menzies School of Health Research, Charles Darwin University, Darwin, NT, Australia
[9] School of Computing and Information Systems, Melbourne School of Engineering, University of Melbourne, Melbourne, VIC, Australia

Corresponding author
Rebecca H. Chisholm,
r.chisholm@latrobe.edu.au

## ABSTRACT

Households are known to be high-risk locations for the transmission of communicable diseases. Numerous modelling studies have demonstrated the important role of households in sustaining both communicable diseases outbreaks and endemic transmission, and as the focus for control efforts. However, these studies typically assume that households are associated with a single dwelling and have static membership. This assumption does not appropriately reflect households in some populations, such as those in remote Australian Aboriginal and Torres Strait Islander communities, which can be distributed across more than one physical dwelling, leading to the occupancy of individual dwellings changing rapidly over time. In this study, we developed an individual-based model of an infectious disease outbreak in communities with demographic and household structure reflective of a remote Australian Aboriginal community. We used the model to compare the dynamics of unmitigated outbreaks, and outbreaks constrained by a household-focused prophylaxis intervention, in communities exhibiting fluid vs. stable dwelling occupancy. We found that fluid dwelling occupancy can lead to larger and faster outbreaks in modelled scenarios, and may interfere with the effectiveness of household-focused interventions. Our findings suggest that while short-term restrictions on movement between dwellings may be beneficial during outbreaks, in the longer-term, strategies focused on reducing household crowding may be a more

effective way to reduce the risk of severe outbreaks occurring in populations with fluid dwelling occupancy.

# INTRODUCTION

For many infectious diseases, it is assumed that the risk of transmission within households exceeds that in the wider community due to the increased opportunity they provide for repeated and prolonged close contact between the people who live in them (*Goeyvaerts et al., 2018*; *Endo et al., 2019*). Due to this increased risk, households are often the focus of infectious disease control strategies. For example, household contacts of invasive Group A *Streptococcus* cases are estimated to have a 2,000-fold increased risk of developing the disease themselves (*Oliver et al., 2019*). For Meningococcal disease, the equivalent increase in risk is estimated to be between 500–800-times (*De Wals et al., 1981*). As such, prophylaxis of household contacts of cases for both of these infectious diseases is recommended to prevent further spread (*Oliver et al., 2019*; *De Wals et al., 1981*).

Much of our understanding of household structure, and hence its representation in mathematical models of disease transmission comes from descriptions of census data. However, these descriptions frequently rely on the notion of a stable 'nuclear household' (i.e. comprising two parents and their children). This notion may fail to capture the complexities and nuances of populations with very different household structure and dynamics. In many settings, households differ in their composition—the people they contain and their relationships to each other. Households may contain extended family members, multiple family units, and unrelated people. For example, in Thailand, the proportion of households *not* considered to be 'nuclear' is estimated at close to 50% (*Dommaraju & Tan, 2014*). In Vietnam, this proportion is estimated to be one third, the majority of which are so-called 'stem households' which include adults, their parents, and possibly their children (*Dommaraju & Tan, 2014*). The proportion of households where there is co-residence of children under 15 years of age with older people over 60 differs greatly throughout the world—in Senegal it is 37% , but just 0.2% in the Netherlands (*United Nations, Department of Economic & Social Affairs, Population Division, 2017*).

Patterns of membership of households may also vary over time. People may spend time in multiple housing units, blurring the relationship between the household as a unit of social organisation and the physical dwelling (*Smith, 1992*). For example a study in Northern Malawi found that households were distributed across between one and twelve dwellings (mean of 1.7 dwellings per household), with between one and nineteen persons occupying each dwelling per night (mean 3.0) (*Fine et al., 1997*). Australian Aboriginal and Torres Strait Islander households can also be distributed across more than one physical dwelling. One study of the occupancy of a single dwelling in a remote Australian Aboriginal community over time revealed that in addition to core residents, the dwelling

was also regularly occupied (although less frequently) by an extended household compromising other relatives and close associates (*Musharbash, 2008*). Over the course of just over a year, more than 100 unique people were observed to stay at the dwelling for at least one night. The flux in occupancy of individual dwellings potentially results in an increased risk of introduction into dwellings, and a continually changing population at risk of household-level infection transmission, particularly if there is also high rates of overcrowding (27.3% of Indigenous Australians living in remote communities live in households requiring at least one additional bedroom, based on the Canadian National Occupancy Standard for Housing Appropriateness, compared to 5.5% for non-Indigenous Australians, *Australian Institute of Health and Welfare (2017)*). The implications of this type of fluid dwelling occupancy on infectious disease transmission and control are unknown.

In this study, we introduce an individual-based model incorporating a more flexible representation of household membership distributed across multiple dwellings. We calibrate our model to a remote Australian Aboriginal community to capture observed demographic, household and mobility characteristics of the population. We then use the model to simulate unmitigated and mitigated (through a household-focused prophylaxis intervention) outbreaks of an influenza-like illness where the risk of infection transmission between contacts residing in the same dwelling is greater than those in the wider community. Model outputs are compared to those from a more traditional household model assuming stable dwelling occupancy, to quantify the impact of distributed households and fluid dwelling occupancy on the dynamics and control of communicable diseases outbreaks.

# MATERIALS AND METHODS

## Individual-based model of population and infection dynamics

### Population structure

Our individual-based model tracks the age and current residence of individuals in a community over time. The community is comprised of $N$ individuals and $H$ physical dwellings. An individual's age is updated each day, and individuals are lost due to natural death at an age-dependent rate. When a death occurs, a new individual aged zero is born into the population so that the population size $N$ is constant.

### Population mobility

The mobility model is based on the Australian Indigenous mobility framework proposed in (*Musharbash, 2008*). This study tracked the number of people that stayed at least one night in a particular dwelling in the remote Australian Aboriginal community, Yuendumu during the 221 nights for which this data was recorded (these 221 nights were not continuous, but occurred during the 467 day study period). The cumulative number of nights stayed by each person was reported. The authors identified four types of residents, based on the amount of time spent in the dwelling: so called core residents, who were present 60–100% of the time, regular residents, who were present 20–34% of the time, other residents who stayed less frequently on an on-and-off basis and were

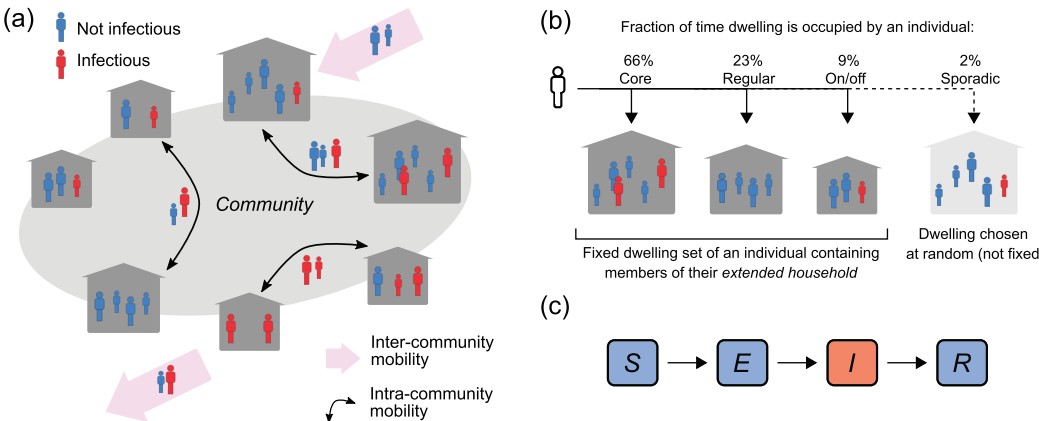

**Figure 1 Population, mobility and infection model.** (A) Intra-and inter-community mobility results in the movement of infectious ($I$) and non-infectious individuals ($S, E, R$) within and between communities. (B) Individuals identify with three dwellings in their community: their core residence, where they spend most nights, a regularly-visited residence, and an on-off residence, where they stay less frequently on an on-and-off basis. Individuals may also sporadically stay in a dwelling chosen uniformly at random from all dwellings in the community. (C) *SEIR* outbreak model. Individuals are born susceptible to infection, can become exposed to infection through contact with an infectious person, before progressing to infectiousness, and then become immune to re-infection following recovery from infection.

present 4–16% of the time, and many sporadic short-term visitors who stayed for between 1 and 6 nights.

In line with this framework, individuals in our model are assigned (uniformly at random) to a fixed set of three dwellings within their community, which we assume remain fixed over the time frames we are considering in this study (less than one year), and which we refer to as their *dwelling set*. These dwellings represent their core residence, where they spend most nights, a regularly-visited residence, and an on-off residence, where they stay less frequently on an on-and-off basis (see Fig. 1). We refer to individuals who have the same core residence as a *core household*, while an individual's *extended household* consists of all core, regular and on-off residents of their dwelling set.

An individual's *current residence* can change due to population mobility. We have two types of mobility in the model: between-dwelling mobility (intra-community); and between-community mobility (inter-community).

Within the community, each day, an individual's current residence is chosen to be either their core, regular or on/off residence with respective probabilities $p_c$, $p_r$ and $p_o$, where $p_c > p_r > p_o$. There is also a small probability $p_s$, where $p_s < p_o$ and $\sum_i p_i = 1$, that their current residence will be a dwelling chosen uniformly at random.

To capture inter-community mobility, each day, $A$ individuals (where $A$ is a Poisson distributed random variable with mean $\alpha N$, and $\alpha$ is the mean per capita migration rate) are chosen uniformly at random to be replaced by immigrants (thus ensuring that community size remains constant). Immigrants are assumed to have a similar age to individuals in the population. This is implemented by specifying that an immigrant will have the same age as an individual selected uniformly at random from the population. Immigrants are assigned a current residence chosen uniformly at random from all
dwellings in the community, as are the dwellings which make up their dwelling set (their core, regular and on/off residences). All immigrants are assumed to be susceptible to infection.

In an extended version of the model (described in the Supplemental Material) we consider an additional type of mobility—the regular influx of temporary visitors into the community due to two types of events: funerals (which take place after the death of a community resident) and reoccurring events, such as sporting matches or festivals. This type of mobility leads to temporary changes in the community size.

### Infection dynamics

We use an *SEIR* (Susceptible-Exposed-Infectious-Recovered) transmission model to simulate an outbreak of an influenza-like illness in the community (see Fig. 1C). Individuals in the community are classified according to their infection status: they are either susceptible to infection (i.e. they can acquire the infection from an infectious contact), exposed (i.e. they have a latent infection and are not infectious), infectious (i.e. they have an active infection and can infect susceptible contacts), or recovered (i.e. they have recovered from the infection and are protected from re-infection). This infection status can change over time due to a transmission event, the progression to infectiousness, or due to the clearance of an infection (detailed below).

Each day, individuals with the same current residence make contact with each other (we refer to these contacts as household contacts), and we simulate daily contacts that occur between individuals in the wider community (i.e. between individuals with different current residences, which we refer to as community contacts). These community contacts occur at age-dependent rates $c_{u,v}$, where $c_{u,v}$ is the daily rate of contact of an individual in age-category $u$ with individuals in age category $v$. Community contacts are chosen uniformly at random from the pool of individuals in the relevant age category.

If a susceptible person makes contact with an infected individual with a different current residence, the susceptible person becomes infected (entering the exposed class) with probability $q$. Household contacts (between individuals with the same current residence) are assumed to be more intense than community contacts. We translate this increased intensity into a probability of transmission per contact that is higher by a factor of $\hat{q} \geq 1$ for these household contacts, compared to community contacts. The duration of latent and active infection are assumed to be exponentially distributed with respective mean duration of $1/\sigma$ and $1/\gamma$. Once an individual clears an infection (and enters the recovered class), they can no longer be infected.

Simulated outbreaks were seeded with one infectious individual (chosen uniformly at random), and with the rest of the population in the susceptible class, and were run until the end of the outbreak (when there were zero infected individuals left in the population).

### Dynamics of a core household-focused prophylaxis intervention

Finally, we also consider outbreaks where a prophylaxis intervention is administered to the core household members of an infectious person.

We assume that this prophylaxis intervention is administered once an infected person enters the infectious state. We do not explicitly model the onset of symptoms in the model. However, if symptom onset corresponds to the onset of infectiousness, then the timing of this intervention corresponds to there being no delay in the core household receiving prophylaxis from symptom onset of the index case.

We consider outbreak scenarios where the intervention is 100% and 50% effective at protecting the core household from contracting the infection, if they hadn't been previously infected and/or recovered.

## Model parameterisation and description of outbreak simulation scenarios

We parameterised the model to be consistent with demography and mobility in remote Australian Aboriginal and Torres Strait Islander communities.

We considered outbreaks in communities of size $N = 2,500$ and $N = 500$ individuals, with respective number of dwellings $H = 358$ and $H = 80$, that are reflective of a large and small-medium community in the NT (*Australian Bureau of Statistics, 2018a*). With these values, the mean number of core residents per house was 7 and 6.3, respectively. We also explored scenarios in populations with lower numbers of core residents per house (i.e. with either $(N, H) = (2,500, 833)$ or $(N, H) = (500,160)$, so that the mean number of core residents per house was 3 and 3.1, respectively), to explore the impact of fluid dwelling occupancy in less-crowded communities.

Mortality rates and the initial age distribution were taken from the most recent census data of Aboriginal and Torres Strait Islander Australians in the Northern Territory (NT), Australia (*Australian Bureau of Statistics, 2018b*) (see Fig. 2A). We set the intra-community mobility probabilities to be $(p_c, p_r, p_o, p_s) = (0.66, 0.23, 0.09, 0.02)$ based on data (summarised above) of house occupancy over time in a single household from the remote Australian Aboriginal mobility study in Yuendumu (*Musharbash, 2008*). Inter-community mobility patterns are not described in this setting although, anecdotally, Aboriginal Australians are described as having a higher than average rate of mobility compared to non-Aboriginal Australians (*Morphy, 2007*). We set the per capita expected migration rate $\alpha$ to be between [0.002,0.004] per day, which corresponds to, on average, between [5, 10] migration events per week when the population size $N = 2,500$.

To date, there have been no studies measuring contact patterns outside of households in remote Indigenous Australian communities. Age-dependent contact data that differentiates between household and non-household contacts is available for rural populations in Kenya (*Kiti et al., 2014*), and we used this to specify the age-dependent community contact rates $c_{u,v}$ in our model.

Infection parameters were chosen to be consistent with influenza-like illness: the mean duration of latency $1/\sigma$ was set to between [1, 3] days, as was the duration of infectiousness $1/\gamma$. We do not have data to inform the within-house transmission factor $\hat{q}$. Therefore, we considered two different scenarios: a high household-infection risk scenario where $\hat{q}$ is set between [3, 5], and a medium household-infection risk scenario where $\hat{q}$ is set between

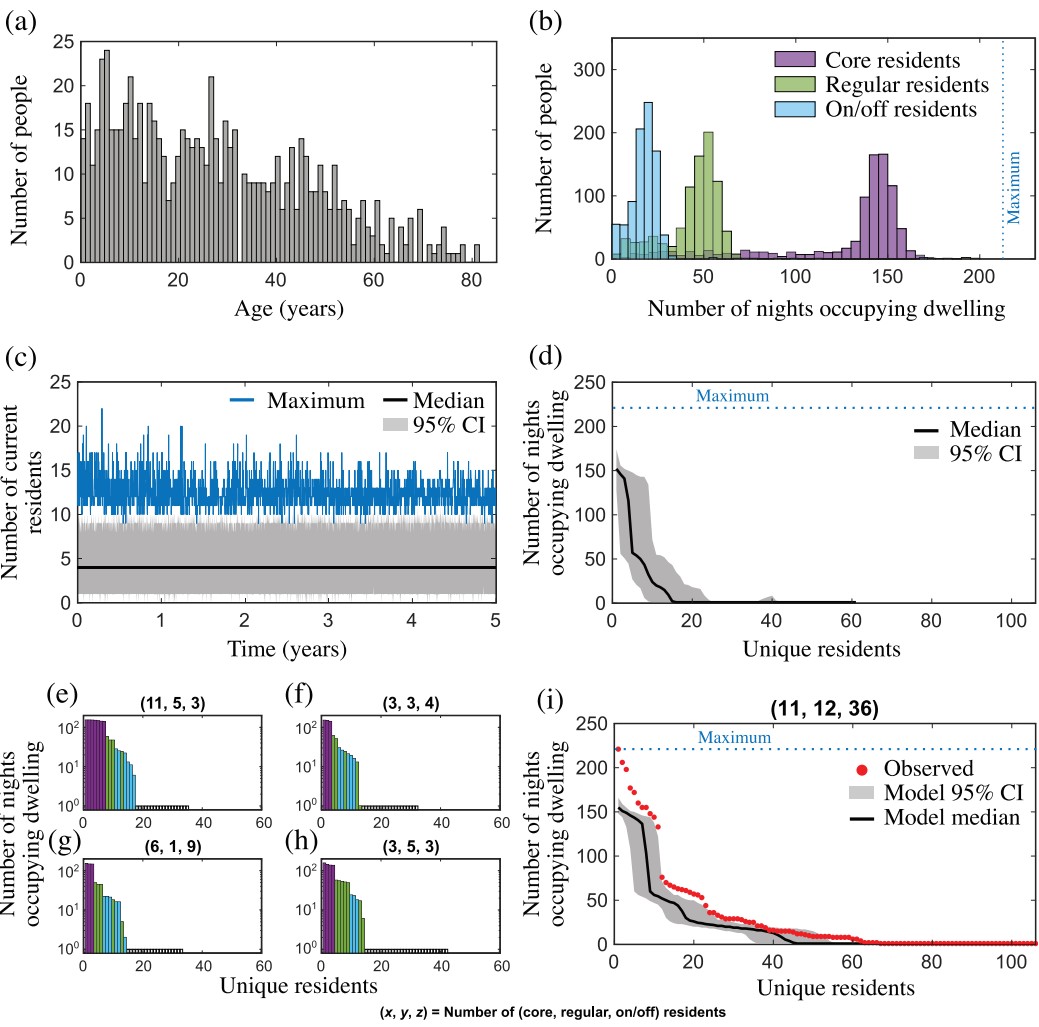

**Figure 2 Population and mobility model outputs from (A–H) one simulation; and (I) 100 simulations, for a community with similar characteristics to Yuendumu (NT, Australia).** (A) Age distribution of the population in years; (B) number of nights that core (purple), regular (green) and on/off (blue) residents occupied their core, regular and on/off dwellings, respectively, over 221 nights; (C) the distribution of the number of current residents in each dwelling over 5 years, showing the median (black line), maximum (blue line) and 95% CI (grey shading); (D) the distribution of cumulative dwelling occupancy over 221 nights for all dwellings. The nth unit of the horizontal axis represents the nth-most regular occupant of a dwelling, and the vertical axis represents the median (black line) and 95% CI (grey shading) for the cumulative number of nights stayed by this occupant; (E–H) the cumulative dwelling occupancy over 221 nights for four exemplar dwellings. Each bar represents a unique individual (coloured according to resident type: core, purple; regular, green; on/off, blue; white, sporadic visitor) who stayed at the dwelling for at least one night, and the height of the bar represents the cumulative number of nights the individual was present (note, a log scale is used). Individuals are shown in order of decreasing occupancy, and the title of each subplot shows the number of (core, regular, on/off) residents for that dwelling at the end of the simulation; (I) observed occupancy (red dots) vs. model occupancy (median and 95% CI from 100 simulations) for a dwelling with 11 core, 12 regular, and 36 on/off residents.

[1, 3]. We set the community transmission probability $q$ to be between [0.002, 0.004] which, in the high household-infection risk scenario, led to outbreaks where greater than 50% of the population became infected, when the outbreaks took off. Results are also
provided in the Supplementary Material where we assumed a higher transmissibility of the infection with $q$ set to be between [0.004, 0.006].

To account for uncertainty in the model parameters, for each population and infection scenario considered, we generated 1,000 samples from the parameter space using Latin Hypercube Sampling (*Blower & Dowlatabadi, 1994*). The parameters $\alpha$, $1/\sigma$, $1/\gamma$, $\hat{q}$, and $q$ were sampled from uniform distributions with upper and lower bounds as described above. All other parameters were held constant.

All outbreak scenarios were re-run in a population assuming stable dwelling occupancy (i.e. with the intra-community mobility probabilities set to $(p_c, p_r, p_o, p_s) = (1, 0, 0, 0)$, and again in populations where the core household-focused prophylaxis interventions, described above, were implemented, to understand the implications of fluid dwelling occupancy on outbreak dynamics and control.

The model is implemented in Matlab and the code needed to regenerate all figures and tables is available at https://github.com/rhchisholm/transmission-complex-households.

## RESULTS

### Population mobility model leads to dwelling occupancy distributions consistent with observations in a remote Australian Aboriginal community

To determine whether our model leads to dwelling occupancy distributions that are consistent with that observed in Yuendumu, we first set up our model population to have similar characteristics to this community. According to the most recent census data, Yuendumu has a population size $N = 759$, an average household size of 4.3 (which we used to estimate the number of dwellings $H = 176$), and people have a median age of 28 (*Australian Bureau of Statistics, 2016*). We then simulated population and mobility dynamics using our model, collecting occupancy data from all dwellings over 221 nights (randomly selected during a 467 day period), and compared this to the occupancy distribution from the Yuendumu study (data was extracted from Figure 10 in *Musharbash (2008)* using the open-source tool, Engauge Digitizer Version 12.1). A sample of these model outputs is shown in Fig. 2. The median of the distribution of the number of current residents over time closely matches the average household size observed in Yuendumu (Fig. 2C), and the maximum occupancy in the model fluctuates between 9 and 22, which is consistent with other studies reporting household size in remote communities (*McDonald et al., 2008*; *Vino et al., 2017*). There are clear steps in the distribution of the cumulative number of nights stayed by different resident types (Figs. 2D–2H), as was observed in the original study (Fig. 2I). We found that the widths of these occupancy steps were a reflection of the number of residents of each type (core, regular, on/off and sporadic visitors) associated with a dwelling, which differed between dwellings (Fig. 2E). The observed cumulative occupancy in the Yuendumu dwelling (*Musharbash, 2008*) largely matched the distribution of model occupancy from a dwelling with the same number of core, regular and on/off residents as this dwelling (Fig. 2I). There was limited overlap of the observed data with the distribution of cumulative occupancy for all houses

**Table 1 Statistics from model scenarios of unmitigated outbreaks in communities of size 2500, including percentage of simulations that led to an outbreak (take off %), and the median (50% CIs) of the outbreak duration and final size.**

| Scenario | Dwelling occupancy | Take off (%) | Duration (days) | Final size |
|---|---|---|---|---|
| Baseline | Fluid | 52.8 | 112 (91, 136) | 1,760 (1,304, 2,040) |
| Baseline | Stable | 52.3 | 129 (105, 160) | 1,494 (1,073, 1,789) |
| Lower $\hat{q}$ | Fluid | 37.7 | 139 (104, 179) | 1,051 (507, 1,414) |
| Lower $\hat{q}$ | Stable | 34.3 | 141 (112, 183) | 995 (473, 1,319) |
| Less crowded | Fluid | 33.8 | 147 (117, 184) | 910 (446, 1,276) |
| Less crowded | Stable | 33 | 140 (88, 181) | 725 (118, 1,070) |

in the population (comparing Figs. 2D and 2I). However, this was expected, given the difference in the number of residents in the Yuendumu dwelling, compared to the population average in the model (which was much lower). The greatest discrepancy between the observed and model occupancy for the single dwelling with the same number of residents related to the most regularly occupying core residents, with the model consistently underestimating the nights stayed by these residents. This was also the case when we considered the extended model with event migration (Fig. S1). Nevertheless, both models qualitatively capture the fluid dwelling occupancy observed in a remote Australian Aboriginal community.

## Fluid dwelling occupancy leads to faster and more-intense outbreaks

We then used our model to simulate outbreaks of an influenza-like illness in communities with a population size and core dwelling size distributions reflective of large and small-medium remote Aboriginal communities in the NT, Australia (the population sizes were 2,500 and 500, and the mean number of core residents per dwelling was 7 and 6.3, respectively) (*Australian Bureau of Statistics, 2018a*). Key model outputs are shown in the main manuscript for large communities, and the analogous outputs for the small-medium communities are provided in the electronic Supplementary Material. All outbreaks were seeded with a single infectious person, and model outputs (summarised in Table 1 and Table S1) were compared to those from equivalent simulations in communities assuming stable dwelling occupancy.

We found that infection introductions were just as likely to lead to outbreaks in communities with fluid dwelling occupancy as they were in communities with stable dwelling occupancy. However, for outbreaks which did take off, those which occurred in communities with fluid dwelling occupancy were consistently more intense than those in communities with stable dwelling occupancy (Fig. 3A). That is, in communities with fluid dwelling occupancy, outbreaks were typically larger in overall size (the total number of people infected during outbreaks), had a higher, and earlier peak (the time in the outbreak when the number of infectious people was highest), and had a shorter duration than those in communities with stable dwelling occupancy.

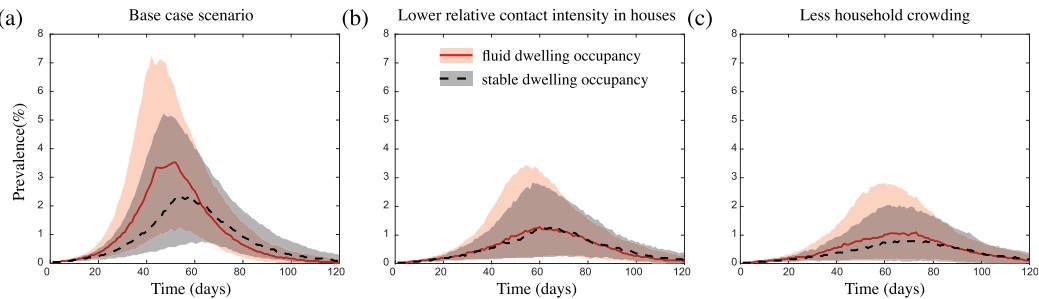

**Figure 3 The impact of fluid dwelling occupancy on influenza-like outbreaks in a population of size N = 2,500 assuming (A) a high-level; (B) a medium-level, of increased risk of transmission from household contacts compared to community contacts; and (C) less crowding in dwellings.** The lines and shading show the median and interquartile ranges of the population prevalence of infection over time when there is fluid dwelling occupancy (red solid line, red shading); compared to when there is stable dwelling occupancy (black dashed line and grey shading).

These differences in outbreak intensity were less noticeable when we considered outbreak scenarios (i) with a lower increased risk of infection transmission between contacts residing in the same dwelling compared to those in the wider community (Fig. 3B; Figs. S11A and S11B); (ii) in communities with a lower extent of household overcrowding (Fig. 3C); and/or (iii) with a more transmissible pathogen (Figs. S2A, S2C, S11A and S11B). These results were robust to the sizes of the communities considered (Fig. S3), and to the inclusion of event-based mobility in the fluid dwelling-occupancy model (Figs. S4 and S5).

## Higher outbreak intensity is driven by an increased number of unique and higher-risk, household contacts

To understand why communities with fluid dwelling occupancy experienced more intense outbreaks, we inspected the number and types of contacts of infectious people over the course of outbreaks (Fig. 4; Fig. S6). We found that the greatest relative difference between the contact patterns of infectious people between the fluid occupancy model (with and without event-based migration) and stable occupancy model was in relation to the number of unique individuals they contacted within dwellings, which was much greater in communities assuming fluid dwelling occupancy compared to stable dwelling occupancy, independent of the community size considered. Neither the number of unique community contacts, nor the total number of contacts of infectious people within or outside of dwellings were as affected by the type of dwelling occupancy model assumed, which suggests that the higher outbreak intensity observed in model communities with fluid vs. stable dwelling occupancy was driven by the increased number of unique, and higher-risk, household contacts.

## Fluid dwelling occupancy decreases the impact of a core household-focused prophylaxis intervention

Finally, we explored the effect of fluid dwelling occupancy on the impact of a core household-focused prophylaxis intervention that could be implemented during outbreaks.

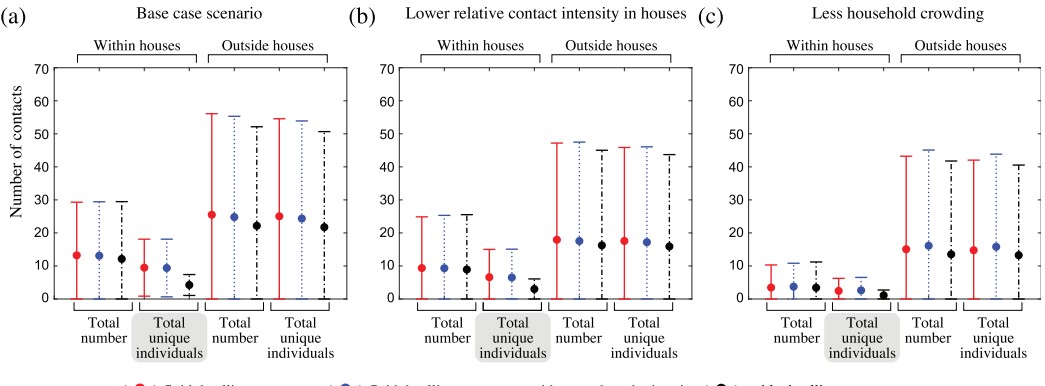

**Figure 4 The impact of fluid dwelling occupancy on the distribution of the number of contacts of infectious people during outbreaks in a population of size $N$ = 2,500 assuming (A) a high-level; (B) a medium-level, of increased risk of transmission from household contacts compared to community contacts; and (C) less crowding in dwellings.** Each disk with error bars shows the mean of means ± one pooled standard deviation of either the total number of contacts, or the total number of unique individuals contacted (as indicated in the plots) during the infectious period of infected individuals, when there is fluid dwelling occupancy (red, solid lines); fluid dwelling occupancy with event-based migration (blue, dotted lines); compared to when there is stable dwelling occupancy (black, dash-dot lines).

This intervention was administered to an infected person's core household at the time of infectiousness onset (which was, on average, between 1 and 3 days post exposure), which protected the core household from contracting the infection, if they hadn't previously been infected and/or recovered.

In all scenarios considered, the intervention reduced outbreak size (Table 2; Tables S2 and S3), although this occurred to a lesser extent in communities with less crowding in dwellings (likely because the average population coverage of the intervention per treated core household was reduced) (Fig. 5; Fig. S7), or when we considered either a more transmissible pathogen, or a pathogen with a higher relative risk of infection transmission within dwellings (Figs. S11C and S11D). In scenarios where we assumed the intervention was 100% effective at protecting a case's core household from contracting the infection, the intervention had a greater impact on outbreak size in communities with stable dwelling occupancy, compared to those with fluid dwelling occupancy (Fig. 5; Fig. S7). In scenarios where we assumed the intervention was 50% effective, there was little to no difference in the impact of the intervention between communities with fluid vs. stable dwelling occupancy, unless household crowding was reduced. In this latter case, the 50% effective intervention had a greater impact in communities with stable vs. fluid dwelling occupancy (Figs. 5C and 5F). Again, these results were robust to the sizes of the communities considered (Fig. S8), and to the inclusion of event-based mobility in the fluid-household membership model (Figs. S9 and S10). In some scenarios where we assumed the intervention was 50% effective, the duration of the outbreak was increased by the intervention, although the total size was reduced (Figs. 5A and 5D; Figs. S7A, S7C, S7D, S7F, S8G, S8I, S8J, S8L, S9A, S9D, S9G, S9I, S9J, S9L and S10G, S10J). This occurred

**Table 2 Statistics from model scenarios of mitigated outbreaks in communities of size 2,500, including the percentage reduction in the median value of the outbreak duration and final size, compared to the equivalent unmitigated scenarios, for 100% and 50% effective interventions.**

| Scenario | Dwelling occupancy | Median duration reduction (%) with intervention effect | | Median final size reduction (%) with intervention effect | |
|---|---|---|---|---|---|
| | | 100% | 50% | 100% | 50% |
| Baseline | Fluid | 29 | −4 | 87 | 48 |
| Baseline | Stable | 55 | −2 | 97 | 49 |
| Lower $\hat{q}$ | Fluid | 45 | 11 | 87 | 61 |
| Lower $\hat{q}$ | Stable | 59 | 10 | 94 | 61 |
| Less crowded | Fluid | 35 | 10 | 77 | 39 |
| Less crowded | Stable | 37 | 13 | 85 | 54 |

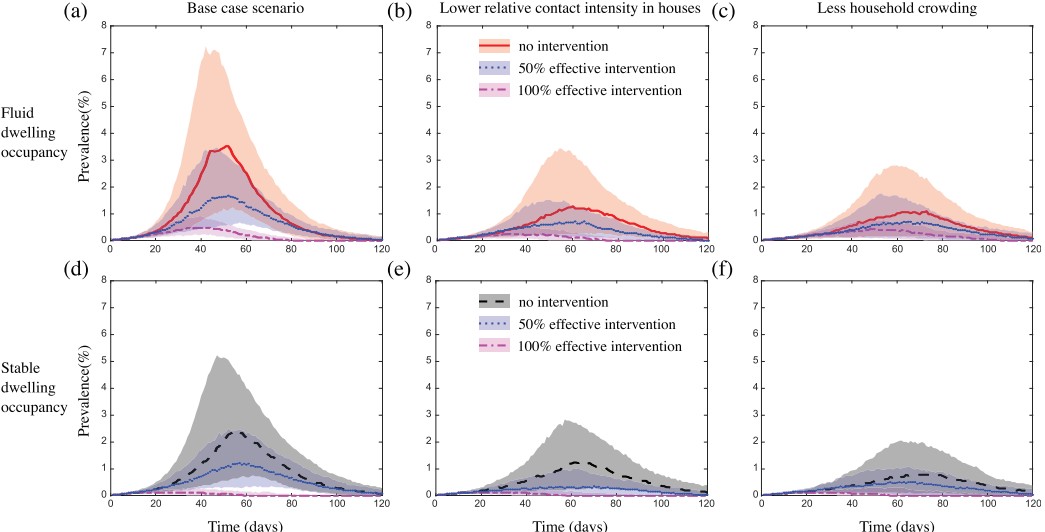

**Figure 5 The impact of fluid dwelling occupancy on the effect of a household-focused prophylaxis intervention that is 50% effective and 100% effective in a population of size $N = 2,500$ assuming (A and D) a high-level; and (B and E) a medium-level, of increased risk of transmission from household contacts compared to community contacts; and (C and F) less crowding in dwellings.** The lines and shading show the median and interquartile ranges of the population prevalence of infection over time when there is (A–C) fluid dwelling occupancy (unmitigated outbreak: red solid line and shading; with 50% effective intervention: blue dotted line and shading; with 100% effective intervention: magenta dash-dot line and shading); compared to when there is (D–F) stable dwelling occupancy (unmitigated outbreak: black dashed line and shading; with 50% effective intervention: blue dotted line and shading; with 100% effective intervention: magenta dash-dot line and shading).

more frequently in communities of size 2,500, and when we considered a more transmissible pathogen.

## DISCUSSION

It is generally assumed that households are associated with a single physical dwelling which is considered to be a high-risk location for the transmission of many infectious diseases. However, the assumption of a one-to-one correspondence between households and

dwellings does not appropriately reflect households in some populations, such as the Australian Aboriginal and Torres Strait Islander communities considered here, where households can extend across multiple physical dwellings leading to fluid groups of people occupying individual dwellings. In this study, we showed that communities made up of such extended households have the potential to experience larger and more intense outbreaks of infectious diseases spread by close contact, particularly when there are high levels of household crowding.

## Outbreaks spread rapidly in communities characterised by fluid dwelling occupancy due to close mixing in and between interconnected households

In our model with fluid dwelling occupancy, the extended household of an individual does not, in general, overlap with that of others in their extended household. Thus, multiple extended households can be connected via shared members, leading to large pools of individuals at greater risk of quickly contracting an infection and spreading it to other extended households, and to faster and larger outbreaks. For pathogens where there is even greater relative risk of infection transmission between household contacts compared to between community contacts, the risk of onward transmission beyond an extended household is amplified further, leading to even larger discrepancies in outbreak intensity between model communities characterised by fluid vs. stable dwelling occupancy.

These reflections also help to explain why smaller discrepancies in outbreak intensity were observed between communities with different dwelling occupancy models when either household crowding was reduced or a more-transmissible pathogen was considered. In both of these scenarios, the lower-risk community contacts contributed much more to widespread transmission because, in the first scenario, the number of household contacts was significantly reduced, and in the latter scenario, the overall risk of infection transmission from the more-frequent community contacts had increased.

## Implications for infectious disease control

Our findings contribute to the evidence base that supports reducing household overcrowding as an effective strategy to decrease the risk of severe outbreaks in populations with fluid dwelling occupancy (*World Health Organization, 2018*). They also highlight the limitations of household-focused interventions in these settings, which suggests that such interventions should be scaled up to reflect the interconnectedness of households. Our findings also suggest that an intervention that reduces the number of unique household contacts during an outbreak by, for example, limiting the amount of movement between dwellings, may reduce outbreak intensity for certain pathogens. Further work could explore the effectiveness of such interventions.

## Model limitations

Our study of the impact of a household-focused intervention considered scenarios where the intervention could be implemented at the time of infectiousness onset (on average 1–3 days post exposure). This may not be possible for Australian Aboriginal

and Torres Strait Islander people living in remote communities, where access to health care services can be more challenging compared to people living in regional areas or major cities (*Australian Institute of Health and Welfare, 2018*). Given the higher intensity of outbreaks in communities with fluid vs. stable dwelling occupancy, we expect that longer delays in implementation would further reduce the ability of household-focused interventions to constrain outbreaks in these settings.

The mechanistic model of intra-community mobility proposed in this study was based on data describing the cumulative occupancy over a period of time of a single dwelling in one remote Australian Aboriginal community (*Musharbash, 2008*). While the occupancy distributions generated from our model do resemble this data, it remains an open question whether our model is an accurate reflection of the mechanisms which led to these cumulative patterns. It is also an open question how generalisable this model is to other dwellings in the same community in which the data was collected, to other remote Australian Aboriginal and Torres Strait Islander communities, and to other population settings where households are distributed across multiple dwellings. Longitudinal data of intra-community mobility from multiple dwellings, in multiple communities, and from different populations could help to inform these open questions.

## CONCLUSIONS

Our study highlights why accounting for correct household structure and dynamics in models of infectious diseases that spread through close contacts can be important when analysing outbreaks and the effects of interventions. Our analysis suggests that in populations with fluid dwelling occupancy, short-term restrictions on movement between dwellings may be beneficial during outbreaks, and possibly improve the effectiveness of household-focused prophylaxis interventions. However, in the longer-term, pre-emptive strategies focused on reducing household crowding may be a more effective way to reduce the risk of severe outbreaks occurring in such populations. Pathogens which do not spread via close contacts, for example, those which spread via vectors or which are sexually transmitted, may not necessarily have different outbreak dynamics and responses to interventions in communities with fluid vs. stable dwelling occupancy. Further work could explore the implications of complex household structure and mobility for such pathogens, as well as those which are endemic in populations.

## ACKNOWLEDGEMENTS

We thank all participants of the public health consultation workshop: Community-guided and evidence-based strategies to reduce the burden of *Strep A* in Australian Indigenous communities, held in Darwin Australia, February 2020.

### Funding

This work was supported by seed-funding from a NHMRC programme grant (1131932) and a NHMRC Centre of Research Excellence (APP1058804). There was no additional

external funding received for this study The funders had no role in study design, data collection and analysis, decision to publish, or preparation of the manuscript.

### Grant Disclosures

The following grant information was disclosed by the authors:
Seed-funding from a NHMRC programme grant: 1131932.
NHMRC Centre of Research Excellence: APP1058804.

### Competing Interests

Steven Y.C. Tong is an an Academic Editor for PeerJ. All other authors declare that they have no competing interests.

### Author Contributions

- Rebecca H. Chisholm conceived and designed the experiments, performed the experiments, analysed the data, prepared figures and/or tables, authored or reviewed drafts of the paper, and approved the final draft.
- Bradley Crammond conceived and designed the experiments, authored or reviewed drafts of the paper, and approved the final draft.
- Yue Wu conceived and designed the experiments, authored or reviewed drafts of the paper, and approved the final draft.
- Asha C. Bowen conceived and designed the experiments, authored or reviewed drafts of the paper, and approved the final draft.
- Patricia T. Campbell conceived and designed the experiments, authored or reviewed drafts of the paper, and approved the final draft.
- Steven Y.C. Tong conceived and designed the experiments, authored or reviewed drafts of the paper, and approved the final draft.
- Jodie McVernon conceived and designed the experiments, authored or reviewed drafts of the paper, and approved the final draft.
- Nicholas Geard conceived and designed the experiments, authored or reviewed drafts of the paper, and approved the final draft.

### Data Availability

The code to generate all figures and data is available from GitHub: https://github.com/rhchisholm/transmission-complex-households.

### Supplemental Information

Supplemental information for this article can be found online at http://dx.doi.org/10.7717/peerj.10203#supplemental-information.

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
