# Peer review of "A model of population dynamics with complex household structure and mobility: implications for transmission and control of communicable diseases"

_PeerJ, doi:10.7717/peerj.10203_

## Round 0.1 · original submission · Major Revisions

Please address the reviewer comments on a point by point basis and submit a revised manuscript. We look forward to reviewing your revised document.

Reviewer 1 ·

Basic reporting

The manuscript is well organized and the methods and results are presented in sufficient detail. However, the presentation of some Figures could be improved (see minor comments below)

Experimental design

The study aims to account for fluid household structure, which is an important topic and of epidemiological relevance. The methods are sufficiently described with the replication code reposited online. Overall, the model structure looks plausible and well executed. I only have a few suggestions that the authors might wish to explore as additional/sensitivity analyses.
- Dwelling occupancy was randomly chosen according to the set of probability p_i’s, which means some dwellings may have an extraordinary number of occupants due to randomness. Hoewver, this might not happen in reality because people may avoid already occupied dwellings or make arrangements when to use them. In that case, the number of occupants in each dwelling would be more evened out than under the random-allocation assumption.
- As the authors adopted LHS, it may be informative to show correlations between the parameter values and the outcome (final size/effectiveness) to assess the sensivity of the results to the parameter assumptions.

Validity of the findings

Conclusions seem fair and valid. Analysis was thorough and looks robust enough, although I suggest some additional sensitivity analysis as above.

Additional comments

Minor comments:
- L79: please add context of citation. Does Morphy 2007 suggest there is overcrowding (what is definition?) in Aboriginal population?
- L172-173: I think the representation using a set notation {} here is unclear (a set is usually not ordered). How about explicitly presenting scenarios as a combination of N and H, for example, (N, H) = (2500, 358), (500, 80),… ?
- L182: ditto; () instead of {}.
- Fig2: The readers cannot assess if this is consistent with the observed data in Musharbash 2008. Could there be a comparison between simulated and observed distributions (e.g., bars vs red lines)?
- Fig2c: The x-scales seem random: I would suggest to use the consistent scale as much as possible for comparability.
- Fig5: The 4 curves and shades overlapping each other makes the figure very difficult to read. Could there be any alternative way, e.g., separating fluid and stable occupancy curves in different panels?
- L269-271, 274-276: I would say there was a slight difference with 100% effectiveness and not difference with 50% for Baseline and Lower q scenarios according to Table 2. Also, 100% effectiveness may be rather unrealistic given that timely prophylaxis at time of infection of an idex case would be rarely possible, especially when the model considers a remote population where medical resource might not be fully available.

·

Basic reporting

The article passes on these counts, although I have concerns about the accuracy of figures 2 and figure S1, outlined below. Some minor typographical errors have also been identified in the general comments.

An improvement is recommended for figures 2c, S1c . The plot titles indicate the final numbers of core, regular, and on/off residents, but the distribution of cumulative nights stayed for each class would be more interesting. Since each bar represents a unique individual, I suggest colour-coding and grouping according to residential class within each plot.

Experimental design

The paper passes these aspects.

Validity of the findings

Findings appear sound except for figure 2 and figure S1.
1. Figures 2a and 2b seem identical to figures S1a and S1b, respectively. This is strange since figures 2c and S1c are mutually distinct although still very similar. If this is accurate then it requires comment or explanation, and if it is to be expected from the model then displaying figures S1a, S1b is redundant and they should be removed.
2. Similarly, the plot titles in figures 2c and S1c are surprisingly matched even though the plots are not quite. This requires explanation or comment.


In all other regards the paper meets these criteria.

Additional comments

I also make these suggestions/corrections:
1. The tick-labels on the y-axes vary inexplicably between plots in figure S1c. Every multiple of 50 should be indicated on every y-axis, as they are in figure 2c.
2. Page 8/11, line 275, “protected” should be “protecting”, “the core household of a case” would read better as “a case’s core household”.
3. Page 10/11, line 13, “finding” should be “findings”.

---

## Round 0.2 · accepted · Accept

Thanks for addressing the issues raised by the reviewers and making the necessary changes. Your manuscript is much improved.

Reviewer 1 ·

Basic reporting

No comment

Experimental design

No comment

Validity of the findings

No comment

Additional comments

My comments have been fully addressed and I believe the manuscript is ready for publication.

·

Basic reporting

No comment

Experimental design

No comment

Validity of the findings

No comment

Additional comments

The paper is in a much better state and is much more readable.
It is ready for publication after a missing space is inserted before CI in "95%CI" on line 9 of the caption for figure 2.